# Long Non-Coding RNA Epigenetics

**DOI:** 10.3390/ijms22116166

**Published:** 2021-06-07

**Authors:** Marek Kazimierczyk, Jan Wrzesinski

**Affiliations:** Institute of Bioorganic Chemistry, Polish Academy of Sciences, Noskowskiego 12/14, 61-704 Poznań, Poland; mkazimierczyk@man.poznan.pl

**Keywords:** lncRNA epigenetics, RNA modyfiing enzymes, detection modified nucleotides

## Abstract

Long noncoding RNAs exceeding a length of 200 nucleotides play an important role in ensuring cell functions and proper organism development by interacting with cellular compounds such as miRNA, mRNA, DNA and proteins. However, there is an additional level of lncRNA regulation, called lncRNA epigenetics, in gene expression control. In this review, we describe the most common modified nucleosides found in lncRNA, 6-methyladenosine, 5-methylcytidine, pseudouridine and inosine. The biosynthetic pathways of these nucleosides modified by the writer, eraser and reader enzymes are important to understanding these processes. The characteristics of the individual methylases, pseudouridine synthases and adenine–inosine editing enzymes and the methods of lncRNA epigenetics for the detection of modified nucleosides, as well as the advantages and disadvantages of these methods, are discussed in detail. The final sections are devoted to the role of modifications in the most abundant lncRNAs and their functions in pathogenic processes.

## 1. General Remarks

Although almost 80% of the human genome has been transcribed, only 2% of it (mRNA) codes proteins [1,2]. All of the remaining RNAs belong to a vast group of noncoding RNAs (ncRNAs). Based on their functions, ncRNAs can be classified as housekeeping RNAs or regulatory RNAs [3,4]. Housekeeping ncRNAs, including transfer RNAs (tRNAs), small nuclear RNAs (snRNAs), small nucleolar RNAs (snoRNAs) and ribosomal RNAs (rRNAs), are commonly expressed constitutively. Regulatory RNAs are a type of ncRNA with a strong regulatory impact on the expression of protein-coding genes. Based on their size, regulatory RNAs can be divided into two groups: small noncoding RNAs (sncRNAs), which are miRNAs and piRNAs, and long noncoding RNAs (lncRNAs). LncRNAs, whose sizes range from several hundred to several thousand nucleotides, are structurally similar to mRNAs [5]. Just like mRNAs, lncRNAs are transcribed by RNA polymerase II, and are likewise capped, often spliced and polyadenylated [6,7]. Despite their large size, lncRNAs do not code protein. A number of lncRNAs contain short ORFs, with fewer than 100 amino acids, which code small proteins or micropeptides, [8,9,10]. Many lncRNAs are implicated in gene-regulatory roles such as chromosome dosage compensation, imprinting, transcription, translation, splicing, cell cycle control, epigenetic regulation, nuclear and cytoplasmic trafficking, and cell differentiation [11,12,13]. Recent studies have shown that lncRNAs, like mRNAs, contain modified nucleotides, which regulate cellular activity [14,15]. 

It has been known for many years that modified deoxynucleotides (m^5^C, m^6^A) are present in DNA [16,17]. DNA modifications, which do not alter the DNA’s sequence, but have an impact on gene activity, are known as epigenetic modifications [18,19]. Similarly, the analysis of RNA post-transcriptional modifications revealed that they occur in every living organism. Such changes are called RNA epigenetics or epitranscriptomics. Over 170 RNA chemical modifications have been found in different organisms to date and over 60 RNA modifications have been identified in eukaryotes Modomics database: http://modomics.genesilico.pl/, accessed on 2 November 2017 [20]. There are different RNA molecules containing ribonucleotide modifications, such as tRNA, rRNA, mRNA and noncoding RNA. Most of the modifications are found in tRNAs, ranging from adenosine and cytidine methylation to hypermodified nucleosides such as 3-[(2R,3R,4S,5R)-3,4-dihydroxy-5-(hydroxymethyl)oxolan-2-yl]-4,6-dimethylimidazo[1,2-a]purin-9-one (wyosine) [21,22], 2-thio-5-carboxymethyluracyl (s^2^mcm^5^U) [23], N^6^-isopentenyl adenine (i^6^A) [24] or N6-threonylcarbamoyladenosine (t^6^A) [25], and are not present in other RNA molecules. However, the most abundant modified nucleotides involving lncRNA are methyl nucleotide derivatives, including 5-methyl–cytidine (m^5^C) and 6-methyl–adenosine (m^6^A) (Figure 1) [26,27]. Modified nucleotides, such as pseudouridine and inosine, constitute a separate group [28,29].

Abundant noncoding RNAs, such as transfer RNAs, ribosomal RNAs and spliceosomal RNAs are modified and depend on the modifications for their biogenesis and function. tRNA is one of the most extensively modified RNAs with, on average, 13 modifications per molecule [30]. It is well documented that most tRNA modifications play one of two major roles: (a) stabilizing tRNA’s tertiary structure; and/or (b) aiding in codon–anticodon recognition [31]. Such remarks mainly concern modifications in the D- and T-arms, which stabilize the tRNA structure, while modifications in the anticodon arm affect codon recognition, particularly in the highly modified wobble position [32]. tRNA modification are required for specific aminoacylations by aminoacyl–tRNA synthetases [33]. Yeast tRNA^Glu^, deprived of the hypermodified nucleotide mcms^2^U34, is a poor substrate for GluRS, exhibiting a 100-fold reduction in its specificity constant (kcat/KM). The presence of the m^6^A modification in mRNA regulates several molecular processes, such as transcription, pre-mRNA splicing, mRNA export and stability, as well as translation [34]. While pseudouridine (Ψ) and m^5^C modifications affect mRNA stability [35], mRNA containing Ψ synthetized in vitro was more stable than unmodified RNA with the same nucleotide sequences in mammalian cells. A similar effect was observed when m5C was introduced into mRNA [36]. The knockout of methyltransferase NSUN2 causes a decrease in the expression of p16 mRNA. One of the most prevalent forms of post-transcriptional RNA modifications is the conversion of adenosine nucleosides to inosine (A-to-I), mediated by the adenosine deaminase acting on RNA (ADAR) family of enzymes. A–I editing changes how the codon is read by the ribosome, because I is read as G [37]. Moreover, the presence of inosine in the double-stranded region decreases mRNA stability and alters splice sites. In ribosomal RNAs, the most commonly found modified nucleotides are 2′O methylated nucleotides and Ψ [38]. In *H. sapiens,* 28S RNAs, 10 methylated nucleosides and 95 pseudouridines were found. They are distributed over important regions, including the decoding and tRNA binding sites (the A-, P- and E-sites), the peptidyl-transferase center and the intersubunit interface, facilitating efficient and accurate protein synthesis. Disturbances in tRNA and rRNA methylation processes alter cellular functions and the deregulation of these pathways can lead to complex diseases [39,40].

In this review, we present the characteristics, detection methods and the role of the modified nucleotides present in long noncoding RNAs. We also discuss the influence of lncRNA modifications on the development of mammals and neoplastic processes. 

## 2. RNA Modification Mechanisms

There are three groups of protein involved in modifying RNA metabolism [27,28]. The first group (writers, Figure 2) consists of enzymes introducing modified nucleotides into RNA during posttranscriptional RNA modifications; the second group of proteins interacts with modified nucleotides (readers); and the third group is involved in removing modification labels (erasers).

6-methyl adenosine (m^6^A) is one the most abundant RNA modifications, occurring in divergent members of the RNA family such as mRNA, rRNA, tRNA, snRNA and also lncRNA. It is usually present in several sites per transcript [44]. In humans, the formation of the m^6^A modification is connected with the methylase complex (writer). Core MTase heterodimer complexes comprise of a methyl transferase such as 3 (METTL3) and a methyltransferase, such as 14 (METTL14) [41,45,46,47]. Crystallographic and biochemical studies have shown that METTL3 is S-adenosylmethionine methyltransferase with catalytic properties, while METTL14 serves as an RNA binding platform (Figure 2A) [48]. Additionally, interactions of the METTL3/METTL14 complex with other factors like: the Wilms’ tumor 1-associating protein (WTAP) [49,50,51], KIAA1429 (VIRMA) [45,52], the zinc finger CCCH domain containing protein 13 (ZC3H13) [53], the RNA-binding motif protein 15 (RBM15), and its paralog-RBM15B, have been mentioned (Figure 3) [26,54,55]. The WTAP protein is necessary for METTL3/METTL14 complex activity, because the deletion of this protein decreases the m^6^A modification levels [27,49,56]. Similarly, KIAA KIAA1429 silencing also causes a substantial reduction in the amount of m^6^A in RNA. RBM15/15B interacts with METTL3 in a WTAP-dependent manner and mediates the binding of the complex to specific RNA sites [27,57]. Recently, METTL16 was characterized as another “writer” protein, which interacts with several types of RNAs: mRNA, U6 and lncRNA [58,59,60,61]. Unlike METTL3/METTL14, which modifies A to m^6^A in coding RNAs, METTL16 can methylate both coding and noncoding RNAs [62]. While the METTL3/METTL14 complex plays the role of the writer methyltransferase during RNA modification, METTL16, which is involved in MALAT1 epigenetics, acts as both the m^6^A writer, as well as the reader [63,64].

Like DNA and histone modifications pathways, the m^6^A modification has two specific erasers, FTO and ALKBH5 [64,65]. FTO (fat mass and obesity-associated protein) removes the methylation trace by oxidizing m^6^A to N^6^-hydroxymethyladosine or N^6^-formyladenosine, which are chemically unstable and can hydrolyze to the final adenine product [66,67,68]. Another eraser is the homologous protein ALKBH5, which catalyzes the direct removal of the methyl group from adenine [69]. Both demethylases—namely FTO and ALKBH5—belong to the AlkB family, which contains a conserved iron binding motif and an α-ketoglutarate interaction domain [67,70,71,72]. FTO is highly expressed in the brain, neurons and in muscle [73], while ALKBH5 is prevalently expressed in the testis and lungs [74]. Whereas the FTO protein has been linked to obesity, the ALKBH5 protein is essential to spermatogenesis.

m^6^A modification sites are recognized by “reader” proteins to generate functional signaling. The YT521-B homology (YTH) domain-containing protein family are the most predominant m^6^A readers and directly bind to the m^6^A-modified RNA bases. The human YTH domain containing protein family consists of five proteins, namely, YTHDF1–3 and YTHDC1–2, which are conserved in mammalian genomes [75]. YTHDC1 is a nuclear protein involved in gene splicing [76,77,78]. YTHDF1–3 are putative RNA helicases which, apart from the YTH domain, contain a helicase domain, ankyrin repeats, and a DUF1065 domain [79,80]. Moreover, YTHDC2 acts as a scaffold molecule in regulating the spermatogenesis chromosome silencing effect of lncRNA XIST [53,81]. The eukaryotic initiation factor 3 (eIF3) binds to the m^6^A located in the 5′UTR region of mRNA and is involved in cap-independent translation [82,83,84]. However, members of the hnRNP family, including HNRNPA2B1 and HNRNPC, choose their target transcripts by screening the RNA binding motifs (RBMs), which are more accessible to them as a result of the m^6^A modification [85,86]. This mechanism is known as the “RNA epigenetic m^6^A switch”, which means m^6^A alters the local structure of mRNA or lncRNA, to facilitate the binding of HNRNPs for biological regulation [87]. Other m^6^A readers include the insulin-like growth factor 2 mRNA binding protein (IGF2BP) family, which was reported to regulate the stability of m^6^A methylated RNAs [65,88,89].

**5-methyl cytidine (m^5^C)** was found in both DNA and RNA. Two writer m^5^C methyltransferases (MTases) have been shown to catalyze the m^5^C modification of eukaryotic RNA. However, their substrate specificity and cellular functions are not completely understood [90,91]. Moreover, there are eight known eukaryotic RNA (C5 cytosine) methyltransferases (writers). One of them, RNA DNMT2, resembles DNA methyltransferases in its structure and characteristics [92,93], whereas the second group comprised of seven members, the MTPases (NSUN), contains the conserved NOL1/Nop2/Sun motif [94]. The NSUNs methylate tRNA (NSUN2, NSUN6), rRNA (NSUN1, NSUN5), mRNA (NSUN2), and ncRNA (NSUN2, Figure 2B), as well as mitochondrial rRNA (NSUN4) and mitochondrial tRNA (NSUN3), respectively. NSUN7′s specificity is currently unknown [94]. In 2017, Yang et al. presented evidence through in vitro and in vivo studies that m^5^C formation in mRNAs is mainly catalyzed by the NSUN2 type RNA methyltransferase [95]. NSUN2-mediated m^5^C methylation promotes the export of mRNA from the nucleus to the cytoplasm in an ALYREF-dependent manner. The RNA binding protein, ALYREF is an m^5^C reader and is necessary for the nuclear export of m^5^C-modified mRNAs. Unlike m^6^A demethylases, it lacks an enzyme that specifically converts m^5^C to C. Recently, it has been proven that the ten-eleven translocation (TET 1–3) proteins, a family of 5-methylcytosine dioxygenases, catalyze the successive oxidation of m^5^C to the final product, 5-hydroxymethylcytosine (hm^5^C), and can therefore be classified as eraser proteins [96]. The TET protein oxidation mechanism can be divided into two steps. In the first step, the activation of the dioxide molecule occurs, requiring the presence of Fe (II) ions and α-ketoglutaric acid (αKG), in order to convert the dioxygen molecule into a highly active Fe (IV)-oxo intermediate. In the second step, the C–H bond is oxidized to form a C–OH hydroxy product [97]. Moreover, hm5C modifications of RNA are involved in stem cell pluripotency and impact translation efficiency [97,98]. The TET2 protein binds to the promoter region of the oncogenic long noncoding RNA (lncRNA-ANRIL) and regulates the expression of ANRIL and its downstream genes. Additionally, the overexpression of the TET2 protein inhibits ANRIL lncRNA abundance, resulting in the decreased risk of gastric cancer [99]. However, the function of hydroxymethyl cytosine in other RNAs, including lncRNA, is not fully understood. The oxidation mechanism may provide an additional layer of epigenetic regulation to the mammalian genome. 

**Pseudouridine (****Ψ)** is a ubiquitous modified nucleotide, mainly found in rRNA, tRNA and ncRNA [28]. The estimated cell content is high, exceeding 5%. Ψ has an unusual nucleoside, containing a C–C glycosidic bond, instead of the N-C bond found in the rest of the nucleosides [100]. As a result of U being replaced by Ψ, an additional hydrogen bond donor is present at the non-Watson–Crick edge. The distinct structure of Ψ increases both the rigidity of the phosphodiester backbone, as well as the thermodynamic stability of Ψ–A, compared with U–A [101,102].

Pseudouridine writers, called pseudouridine synthases (PUSs), recognize substrates and catalyze the isomerization of U to Ψ, without the need for cofactors (Figure 2B) [42,100,103]. However, PUS enzymes are unable to isomerize free nucleotides. Pseudourydilation is known to follow two different mechanisms. In the first, the RNA-dependent pathway involves the formation of an RNP complex containing a H/ACA RNA box, cofactors, and four core proteins [104]. Box H/ACA RNAs are among the most evolutionarily conserved families of small ncRNAs and are present in all eukaryotes. In rRNA pseudouridylation, small nucleolar RNAs act as guides that recognize targets with sequence complementarity, thus directing pseudouridylation in a site-specific manner [104,105,106]. In humans, the four core proteins associated with box H/ACA RNAs are CBF5, NHP2, NOP10 and GAR1 [105,107]. However, only CBF5 catalyzes the U-to-Ψ isomerization reaction. In the solved crystal structure, three of the proteins interact with the H/ACA guide RNA or substrate RNA, while GAR1 may regulate substrate loading and release [104,108]. 

Alternatively, the RNA-independent pathway relies upon the direct recognition of targets by PUS complexes, often at conserved structural or sequence motifs [109,110]. In contrast to the m^6^A modification, specific Ψ eraser or reader proteins have not yet been identified. This is due to the high stability of the C–C bond and the inability of the potential eraser proteins to cleave it, what is necessary for them to exchange Ψ for U. Furthermore, the lack of reader proteins that specifically bind to Ψ means it is difficult for proteins to identify the more subtle modification, which results in the C–N bond being replaced by a C–C bond, between uracil and ribose [109]. In humans, 10 proteins (PUS1–10) involved in RNA modification, with an annotated Ψ synthase domain, have been found (writers). These are classified into five families (TruA, TruB, TruD, RluA, and PUS10), based on their bacterial counterparts [108,109,110,111]. Although the primary sequences have diverged, all PUS synthases, including CBF5, share a conserved catalytic domain and likely a conserved catalytic mechanism, based on the solved crystal structure. In the case of m^6^A and m^5^C, the reader and the eraser proteins have been identified [26,27]. It is also possible that there could be readers and erasers for the Ψ modification; however, their existence has not yet been proven.

**Inosine (I)** differs from adenine in that it possesses a carbonyl group instead of an amino group at position 6 of the purine ring. This modification only occurs in the double-stranded regions of mRNA, tRNA, rRNA, and ncRNAs and is catalyzed by the writer protein, adenosine deaminase acting on RNA (ADAR). In vertebrates, a family of three ADAR proteins, ADAR1 (Figure 2C), ADAR2, and ADAR3, has been identified [43,112]. Structural analysis has shown that ADAR enzymes contain a C-terminal conserved catalytic deaminase domain, with two (ADAR2 and ADAR3) or three (ADAR1) dsRBDs in the N-terminal portion. The full length ADAR1 protein also contains a N-terminal Zα domain with a nuclear export signal and a Zβ domain, while ADAR3 has an R-domain. ADAR1 and ADAR2 catalyze all the currently known A-to-I editing sites. In contrast, ADAR3 has no documented deaminase activity. It has been postulated that the heterodimerization of ADAR3 with either ADAR1 or ADAR2 might render ADAR1 and ADAR2 inactive. Inosine essentially mimics the chemical properties of guanosine, therefore ADAR proteins introduce an A-to-G substitution in transcripts. These changes can lead to specific amino acid substitutions, altering protein composition. The presence of inosine in RNA influences mRNA alternative splicing, ncRNA-mediated gene silencing, or changes in the transcript’s localization and stability [113]. The lncRNA-mRNA duplex is recognized by the ADAR double stranded specific enzyme, which converts adenine to inosine [114]. In the case of lncRNA A-to-I editing, most of the information comes from bioinformatics analysis [12]. About 200,000 editing sites exist in human lncRNAs. Most of them (65%) are located within those sites that influence their secondary structure. Accordingly, both edited and unedited lncRNAs can have different functions [115].

## 3. Detection of Modified Nucleotides

The information presented above suggests that lncRNA epigenetics are important to cell differentiation and may be involved in controlling organism development. This has motivated many laboratories to screen ncRNAs for the presence of modified nucleotides, and to study the changes of the modification pattern during cell development [26,27]. 

Application NGS methods to epigenetic mapping is so far difficult because they typically do not detect modified nucleosides [116,117]. Developing single-base resolution sequencing, which could quantify the relatively low abundance of modified nucleotides in lncRNA, is a significant challenge. The identification of transcriptome-wide RNA modifications has been approached using different strategies. 

The study of RNA modification started in 1957 when the first modified nucleoside, pseudouridine, was discovered in bulk yeast RNA, using paper chromatography [118]. The first of these, the direct approach, uses antibody immunoprecipitation. The antibody specifically recognizes the modified ribonucleotides, making it possible to determine modifications at a global level [119,120]. The second direct technique uses two-dimensional thin layer chromatography (TLC) to analyze nucleotide composition [120,121]. During the first step, the isolated RNA molecules are hydrolyzed by nuclease, which leaves 3′ phosphate, then after ^32^P labeling at 5′ side of the nucleotide, 5′3′ diphosphate nucleotide is formed. In the last step, 3′ phosphate is removed and 5′ ^32^P labeled nucleotides are separated by TLC. There are several TLC methods used to detect a specific modification. Liu et al. developed the SCARLET method. The method relies on site-specific cleavage, radiolabeling, followed by ligation-assisted extraction and thin-layer chromatography [122,123]. Using this technique, the exact location of the m6A residue was determined, which are key parameters in studying the cellular dynamics of m6A modification. The SCARLET method starts with total RNA or with a total polyA^+^ RNA sample. In the second step, a candidate site in a candidate RNA of interest has to be chosen. In the third step, RNase H cleavage is guided by a complimentary 2′–OMe/2′-H chimeric oligonucleotide to achieve site-specific cleavage 5′ to the candidate site. The cut site is labeled with ^32^P and the ^32^P labeled RNA fragment is splint ligated to 116-nucleotide single stranded DNA oligonucleotide, using DNA ligase. The sample is then digested with RNase T_1_/A to completely digest all RNA, whereas the ^32^Plabelled candidate site remains with the DNA nucleotide as DNA-^32^P(A/m^6^A)p and DNA-^32^P(A/m^6^A)Cp, which migrate as 117/116 mers on denaturing gel. The labeled band is excised from the gel, digested with nuclease P_1_ into mononucleotides containing 5′ phosphate, and then the m^6^A modification status is determined by TLC [124]. This method was successfully used to determine the modified nucleotides like m^6^A, m^5^C, Ψ, and possibly other unknown modified nucleotides, in several coding and noncoding RNAs. The method is laborious and with low throughput and it may be possible to substitute it with ultra-performance LC–MS methods (UPLC–MS) [124,125].

The high-throughput m^6^A mapping strategies were based on the immunoprecipitation of modified RNA molecules, using m^6^A-specific antibodies coupled to the subsequent NGS sequencing. In the following procedures, m^6^A Seq and MeRIP-Seq, ncRNA is fragmented to a size of about 200 nucleotides and immunoprecipitated by a m^6^A specific antibody, attached to the magnetic beads. Then, the RNA separated with a magnet is subjected to a second round of m^6^A immunoprecipitation. The resulting RNA pool, which is highly enriched with m^6^A-containing RNAs, is used for library construction and NGS sequencing [117,126]. Both methods provide a rather low resolution. In another direct m^6^A detection technique, miCLIP [127,128,129], an additional step was added to the MeRIP-Seq method. The specific antibodies are bound to the m^6^A mark in the RNA chain and cross-linked using UV light, with a wavelength of 254 nm. miCLIP allows for a high-resolution detection of m^6^A in RNAs.

The detection of the m^6^A nucleotide in RNA, using an indirect approach, is difficult because few chemical reagents modify the methyl group. However, NOseq, a method for the detection of m^6^A in RNA after chemical deamination by nitrous acid, has recently been introduced [130]. Nitrous acid deaminates adenosines to inosine, while the m6A residue is resistant to such modifications. The application of NGS after modification to detect m6A sites in MALAT1 lncRNA [130].

The indirect approach, in which chemical compounds selectively react with the modified ribonucleotides, was also used during the detection of m^5^C or Ψ in RNAs. Reverse transcription, which is utilized in the next step, enables the detection of the modification sites [119,120]. The most common indirect method used to determine m^5^C modification sites in RNA, i.e., the bisulfite conversion of cytidine to uridine has been applied successfully to determine the presence of this modified nucleotide in DNA [131]. The method is based on the fact that m^5^C modified cytosine is resistant to bisulfite cytosine deamination. Thus, simply comparing the NGS sequences of RNA molecules subjected to bisulfite treatment with those that remain untreated, should pinpoint the modification sites [132,133,134,135]. There are several biochemical kits on the market, making it possible to prepare m^5^C libraries, which are ready for sequencing. A disadvantage in the detection of m^5^C present in RNA using bisulfite is that a large amount of RNA is initially required to compensate for the high losses caused by this reagent. Moreover, other modifications or double-stranded regions may be resistant to bisulfite treatment, especially under the milder reaction conditions required to maintain RNA integrity. Therefore, this method often requires additional confirmation with the aforementioned direct methods used for the detection of m^6^A modifications—MeRIP and miCLIP. They differ in the application of m^5^C-specific antibodies [136].

An alternative approach, which has been termed the “suicide enzyme trap”, has been employed to characterize the substrates of the following m^5^C-methyltransferases (m^5^C-MTases), NSUN2 and NSUN4 [137,138]. By mutating m^5^C-MTases to form irreversible covalent bonds with target residues, the resulting stable enzyme–RNA complexes are suitable for immunoprecipitation and mapping. This is also the case with the AZA-seq methodology, in which the “suicide inhibitor” nucleotide analog of 5-azacytidine is incorporated into cellular RNA and “traps” m^5^C-MTases for pulldown and sequencing [139].

Determining the Ψ sites in the RNA chain requires indirectly analyzing this modification using carbodiimide chemistry. The chemical typically used for this is 1-cyclohexyl-(2-morpholinoethyl) carbodiimide metho-p-toluene sulfonate (CMCT), which reacts with guanosine, uridine and pseudouridine nucleosides. However, in alkaline conditions (pH 10.4), only Ψ–CMCT adducts are stable [140,141]. The bulky CMCT group attached to N3 on Ψ hinders reverse transcription and results in cDNA being truncated. This facilitates the detection of Ψ at a single nucleotide resolution level [140]. A new version of this method, called CeU-seq or Pseudo seq (N3-CMCT-enriched pseudouridine sequencing), was recently developed [142]. The RNA molecules were incubated with a CMCT derivative, and subsequently using the click chemistry approach, the N3-CMCT-Ψ adduct is labeled with a DBCO–(PEG) 4-biotin complex. The immunoprecipitation of Ψ RNA using streptavidin beads results in the enrichment of Ψ-containing RNA. In the next step, the Ψ enriched RNA is used for cDNA library preparation. In the final step, the library is deep-sequenced using the NGS protocol. Sites of pseudouridylation with single nucleotide resolution can be identified by subjecting the data obtained through NGS sequencing of Pseudo-seq libraries, to computational analysis.

Recently, Pan et al. developed a method that uses a CMC-Ψ-induced RT stop with an additional step of site-specific ligation, followed by PCR, to generate two unique PCR products, that correspond to the modified and unmodified uridine. The modification is visualized in the PCR products using gel electrophoresis [143].

The editing events are typically identified using the direct detection method, by comparing the cDNA sequences with the corresponding genomic DNA sequence [144]. The edited inosine base pairs with cytidine in cDNA, hence editing is visible as an A-to-G sequence change. Recently, sophisticated bioinformatics tools have been developed to maximize detection accuracy, while minimizing the detection of false positives [145,146]. Furthermore, considerable effort has been made to comprehensively detect and remove known SNPs from editing datasets and databases [147,148]. As identifying true editing sites from transcriptome sequencing data is difficult, alternative methods aimed at marking inosine have been developed. Another direct method used to detect editing sites utilizes the co-immunoprecipitation of ADAR enzymes with the bound substrate RNA and the subsequent microarray analysis of these associated RNA sequences [149]. The disadvantage of this method is the fact that the association of ADAR with RNA is not necessarily indicative of editing. An alternative method, based on the finding that glyoxal reacts with guanosine to form a stable adduct, whereas inosine glyoxal adducts are unstable, has been developed. Moreover, guanosine glyoxal/borate adducts are resistant to RNase T_1_ digestion [150,151]. RNase T_1_ specifically cleaves RNA after guanosine or inosine but is inhibited by guanosine glyoxal/borate adducts. The cleavage of glyoxal-modified RNA creates RNA fragments that carry inosine at their termini, as an input for sequencing. A method similar to the direct approach, used to determine inosine modifications, called inosine chemical erasing (ICE), was developed. [152,153]. ICE involves the treatment of RNA with acrylonitrile, which converts the inosine to N1-cyanoethylinosine in the process of cyanoethylation, and results in the formation of an inosine/acrylonitrile adduct that inhibits base pairing with cytidine and stalls reverse transcription. The total RNA is either treated with acrylonitrile or left untreated and then reverse transcribed into cDNA. In untreated RNA, the A or I at a given position is converted into T or G, respectively. In the treated sample, A is converted to T, while the presence of inosine/acrylonitrile adducts blocks reverse transcription, leading to shorter cDNAs. The ICE method was combined with NGS (ICE-seq), requiring the fragmentation of poly(A)-enriched RNA before cyanoethylation and reverse transcription. ICE-seq makes it possible to identify the editing sites throughout the transcriptome. The gel purification of longer cDNA fragments effectively erases these shorter inosine/acrylonitrile adduct-containing cDNAs from the library. The subsequent sequencing and comparison of the libraries identifies inosines, by detecting erased reads upon cyanoethylation [154].

NGS methods based on short-read sequencing have difficulty in determining the modification patterns of the entire transcript sequence. Recently, a direct modification detection method, called nanopore sequencing, has been developed [155,156,157]. The Oxford Nanopore Technologies (ONT) sequencer can directly sequence individual native RNA or DNA molecules. The nanopore sequencer can measure disruptions in the current, compared to the raw current intensity, as the RNA or DNA passes through the nanopore in the dielectric membrane. This technology is able, in principle, to identify the nucleotide passing through the nanopore [158]. Most importantly, the method does not require that the RNA be processed, i.e., converted into cDNA by a reverse transcriptase like other NGS methods, and RNA modifications are therefore preserved. ONT data analysis requires that the use of specialized bioinformatic software and several brands of software, such as Tombo, Epinano or ELIGOS, are available [117,155,159]. Tombo is a software used to detect modifications in DNA and RNA, such as the m^5^C modification in DNA and RNA and the m6A modification in DNA [160]. The EpiNano software is used to detect the m6A modification in RNA [117,155]. The ELIGOS software compares the error profile between native RNA sequences obtained with direct RNA-seq and a reference sequence, which can be in vitro synthesized RNA, cDNA or the RNA background error model [161]. The greatest limitation of the nanopore sequencer is its comparatively low read accuracy, compared with short read sequencers. It needs to be highlighted that direct RNA modification analysis using nanopore sequencing is rapidly developing and becoming more reliable, so its routine application in the field of RNA epigenetics is expected.

## 4. The Impact of lncRNA Epigenetics on lncRNA Function

It is estimated that human cells have over 50,000 lncRNA molecules coded in genes. Some of them contain introns and are spliced by the same machinery as pre-mRNAs [12]. Many mature lncRNAs are modified after transcription (Section 2). The application of NGS methods in combination with bioinformatic analysis revealed the occurrence of several modifications in different types of lncRNA molecules. Integrated data analysis of m^6^A, m^5^C, Ψ and I sequencing studies was performed, highlighted the total amount of modifications in lncRNA, as well as the number of individual lncRNAs, that contain the respective modified nucleotide. The results were as follows: m^6^A—(13357 modifications/12348 lncRNAs); m^5^C—(9965 modifications/1072 lncRNAs); Ψ—(162 modifications/150 lncRNAS); I—(11726 modifications/3374 lncRNAs) [26,144,162]. The role modified nucleosides play in lncRNA is not yet completely understood. Most of the information available concerns the most abundant lncRNAs present in the cell [163]. lncRNAs can be classified as being *cis* acting or *trans* acting [164]. The *cis* acting lncRNAs recruit factors, either to the site of lncRNA transcription or to adjacent loci and involve lncRNA XIST and lncRNA H19 [165,166]. Trans acting lncRNAs act independently of their transcription sites, either by regulating expression from other loci in the nucleus or having transcription-unrelated functions anywhere in the cell involving MALAT1, HOTAIR, and lincRNA1238 [166]. Antisense noncoding RNA in the INK4 locus *ANRIL* is included in *cis* and in *trans* gene regulation [167].

XIST is a 17.5 kb capped, spliced and polyadenylated nucleotide transcript, transcribed from the *XIST* gene and involved in X chromosome inactivation [168,169]. Due to its size, XIST has many modifications like 78 sites m6A. 5 sites m5C and single site of Ψ [26]. In humans, multiple m^6^A sites in XIST repeat regions have been identified. Patil and colleagues have shown that m^6^A formation in XIST, as well as in cellular mRNAs, is mediated by RBM15 and RBM15B, which bind the m^6^A-methylation complex and recruit it to specific sites in RNA [83]. Additionally, the knockdown of RBM15 and RBM15B, or the knockdown of METTL3 methyltransferase, impairs XIST-mediated gene silencing. The depletion of YTHDC1 was shown to result in defective X-chromosome inactivation, whereas tethering YTHDC1 to XIST rescues the phenotype in the absence of a functional m^6^A methylation complex [83]. These data suggest that the biogenesis of m6A and its recognition is required for XIST-mediated transcriptional repression.

Many pathways contribute to the control of gene expression during development. Polycomb repressive complex (PRC2) and XIST are associated with gene repression in various developmental processes, such as X chromosome inactivation and genomic imprinting. 

PRC2 binds with high affinity to the 5′-end region of XIST called the repeat A-region [137]. This functionally important domain is composed of 8.5 repeats of a 26-nucleotide sequence. In human XIST, five m^5^C marks were also detected. The presence of m^5^C methylation in the XIST transcript prevents the binding of the PRC2 in vitro. In Xist lncRNA, the presence of pseudouridylation and A–I editing sites has been confirmed, however, their roles are unknown at this stage so far [170,171]. 

The metastasis-associated lung adenocarcinoma transcript 1 (MALAT1) gene is located at chromosome 11q13 and encodes a 7.9 kb transcript. MALAT1 is a highly modified lncRNA, carrying multiple m^6^A modification sites. Moreover, a modification of the m^6^A site in MALAT1 mediates the m^6^A switch, which allows for the HNRNPC (reader) protein to be bound to a U-rich tract in the strand opposite to the m^6^A modification site [172]. Thus, the m^6^A modification regulates the binding of proteins to MALAT1 lncRNA. The recent analysis of MeRIP seq data revealed thousands of m^6^A switches, which are involved in alternative RNA splicing and abundance [173]. The MALAT1 transcript has a triple-helical RNA stability element at the 3′ end, which can be recognized by METTL16 [61,63]. This suggests the existence of a possible m6 A modification site or a function for guiding METTL16 onto its targets. Recently Wang et al. postulated that m^6^A modifications in MALAT1 are important to the metastatic capacity of esophageal cancer cells, both in vitro and in vivo [174]. They also show that the recognition of the m^6^A mark in MALAT1 by the YTHDC1 reader protein plays a critical role in maintaining the composition of nuclear speckles and their genomic binding sites. Interestingly, the phenotypes induced by MALAT1-m^6^A deficiency could be largely rescued both in vitro and in vivo, by artificially tethering YTHDC1 onto MALAT1. In addition, *MALAT1* lncRNA is subject to post-transcriptional m5C modification; five m^5^C sites been found to regulate chromatin-related roles in other lncRNAs, such as HOTAIR and XIST [135]. Although the exact role played by pseudouridine and inosine modifications remains to be explained, and the presence of each of the three inosines increases the stability of MALAT1 by 2–3 kcal/mol [28,170].

*HOX* transcript antisense RNA (HOTAIR) lncRNA is a transcript of the antisense strand of the *hoxC* gene, which is spliced and polyadenylated [175] Within HOTAIR, 14 individual m^6^A sites were identified, with a single site (A783) being consistently methylated. HOTAIR interacts with the nuclear m^6^A reader YTHDC1 at the methylated A783 and at additional sites [176]. Localization in chromatin strongly depends on the m6A modification at site A783 of HOTAIR, while the modification of other m^6^A sites mediates high HOTAIR levels. Additionally, m^6^A-dependent YTHDC1–HOTAIR interactions are required for gene repression, independent of the expression level and chromatin recruitment. The previous results demonstrate that site-specific cytosine methylation occurs in lncRNA HOTAIR [135]. The methylation of C1683 is widespread in different cell types and it is not limited by the abundance of HOTAIR RNA levels. Furthermore, the degree of methylation of C1683 appears to be remarkably high, suggesting that it might be important to HOTAIR’s functioning. In this respect, it is interesting to note that C1683 is located in the vicinity of the region that has previously been shown to interact with the LSD1 complex [13]. It is therefore tempting to speculate that the methylation of this cytosine may affect the ability of HOTAIR to interact with LSD1.

Yang et al. identified the cytoplasmic long intergenic noncoding RNA 1281 (lincRNA 1281), whose function is regulated by m^6^A modifications [177]. This lincRNA is necessary for the differentiation of mouse embryonic stem cells (mESCs) and acts as a ceRNA by sequestering let-7 miRNAs [177]. Notably, lincRNA 1281 contains m^6^A marks in its 3′-end region, which are required for the binding of let-7 miRNA. It has been proposed that the presence of m^6^A in lincRNA1281 can act as a m^6^A-switch for specific RNA binding proteins, which will eventually regulate their interaction with let-7 miRNA. However, the identity of such proteins has not yet been discovered. A similar mechanism has already been proposed for the binding of the HuR (ELAVL1) protein and miRNAs, to the mRNAs encoding developmental regulators in mESCs.

H19, an imprinted lncRNA with a size of 2.3 kb, plays an important role in embryonic development [178]. The knockdown of METTL3 or METTL14 notably reversed the hypoxic preconditioning-induced (HPC-induced) enhancement of cell viability, anti-apoptosis ability, and lncRN AH19 expression [179]. Methylated RNA immunoprecipitation (IP) indicated that the knockdown of METTL3 or METTL14 decreased the m^6^A levels in lncRNA H19. The RNA binding protein immunoprecipitation (RIP) assay showed that METTL3 and METTL14 can directly bind with lncRNA H19. The m^5^C modification in lncRNAH19 can increase its stability [180]. Furthermore, m^5^C-modified lncRNA H19 can be specifically bound by G3BP1, a well-known oncoprotein, which results in MYC accumulation. NSUN2 was shown to methylate lncRNA H19 and affect its stability. The Ras-GTPase-activating protein-binding protein 1 (G3BP1) was confirmed to bind methylated lncRNAH19, based on the presence of NSUN2. lncRNA H19 has two editing sites, and their presence increases the lncRNA stabilization energy by 3 kcal/mol [171].

The human steroid receptor RNA activator (SRA) is a transcript of the *sra1* gene, whose size ranges from 0.7 to 0.9 kb. It is a dual-function RNA, which acts as both an lncRNA and an mRNA [181]. lncRNA SRA regulates several processes, such as cell cycle proliferation, as well as insulin, Notch, and TNFa signaling [182]. Currently, depending on the technique used, the data show the presence of 1–4 m^6^A modifications in lncRNA SRA, of an unknown function [26]. In a subsequent study, the same authors identified a specific uridine residue in SRA1 (U206), whose modification by PUS1 (or PUS3) might induce a functional switch, which regulates nuclear receptor signaling [26,181]. As of now, inosine has not been detected in this lncRNA.

The human plasmacytoma variant translocation 1 (PVT1 lncRNA) is a large locus, which is longer than 30 kb and is located at 8q24.21. It serves an oncogenic role in a variety of malignant tumors, such as colorectal cancer [183]. PVT1 lncRNA is highly modified and contains m^6^A, m^5^C and Ψ nucleosides [36]. Recently, it has been proven that the m^6^A PVT1 lncRNA modification regulates epidermal stemness via its interaction with the MYC protein [184]. The suggested mechanism involves the association of Pvt1 with the MYC protein and prevents MYC degradation. m^6^A methylation is an important regulator of this process. Moreover, the RNA m^6^A modification mediated by the METTL3/METTL14 complex (Figure 2A) regulates epidermal stemness by controlling Pvt1 and MYC interactions through Pvt1 methylation, uncovering a key and novel molecular mechanism underlying skin tissue homeostasis, regeneration and wound repair. Additionally, PVT1 lncRNA has been identified as a pseudourydilation target [26,27]. Some of the Ψ sites were located within functional lncRNA motifs, indicating the potential regulatory impact of Ψ on lncRNAs.

## 5. Conclusions and Future Directions

Information available in the literature suggests that lncRNA epigenetics play a role in gene regulation through various mechanisms. First, the modification of lncRNAs may change their structure and affect their interactions with proteins, like in the case of lncRNA. Second, the modification of lncRNAs could mediate transcription repression [54]. Third, lncRNA modifications might alter its subcellular distribution [185,186]. Finally, lncRNA modifications regulate the stability of lncRNAs like H19, MALAT1, XIST, etc. However, overall, there are still relatively few studies concerning lncRNA epigenetics. Due to the fact that lncRNA has many distinct functions in the cell, there is a need to further elucidate the components required for lncRNAs modification and recognition. It is likely that the development of new sequencing techniques, such as nanopore sequencing, will facilitate the search for new modified lncRNAs. Modified lncRNAs are involved in oncogenesis [176,187] and they can therefore be excellent biomarkers of neoplastic processes [188]. The underlying mechanisms by which lncRNA modifications contribute to gene regulation and whether and how the RNA epigenetics of mRNAs differs from that of lncRNAs, as well as which proteins are involved in both RNAs remain to be elucidated. Understanding these mechanisms makes it possible to develop lncRNA-targeted therapies.

## Figures and Tables

**Figure 1 ijms-22-06166-f001:**
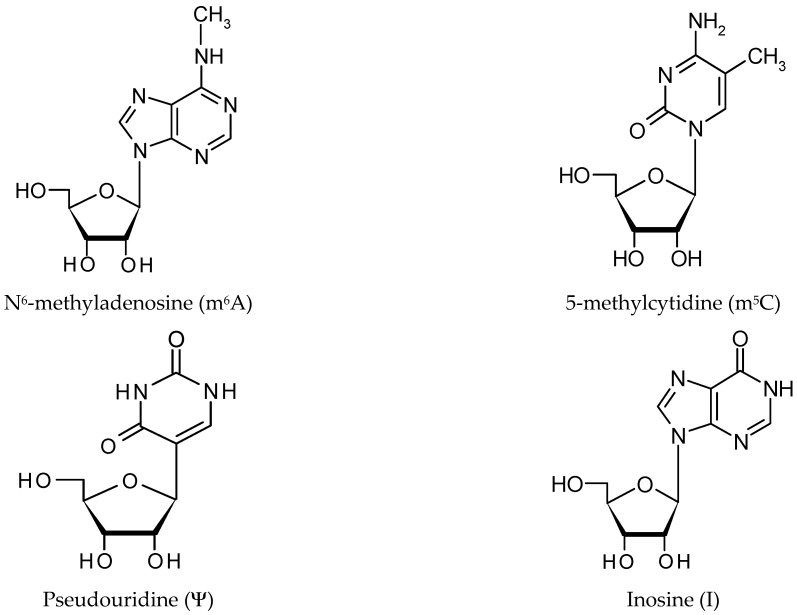
The structure of modified nucleotides found in eukaryotic lncRNA molecules [26,27,28,29].

**Figure 2 ijms-22-06166-f002:**
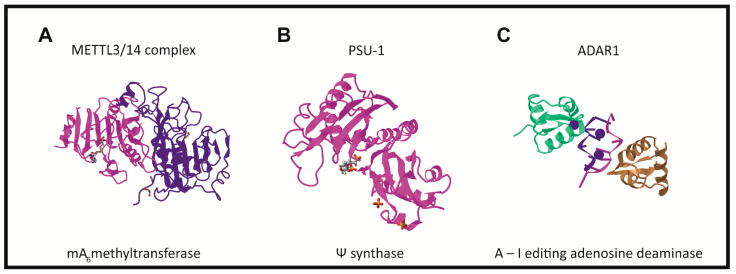
The crystal structure of “writer” enzymes, forming the following lncRNA modifications: (**A**)—METTL3/METTL14 m^6^A methyltransferase [41]; (**B**)—PSU1 pseudouridine synthase [42]; and (**C**)—ADAR1 A-I editing adenosine deaminase [43].

**Figure 3 ijms-22-06166-f003:**
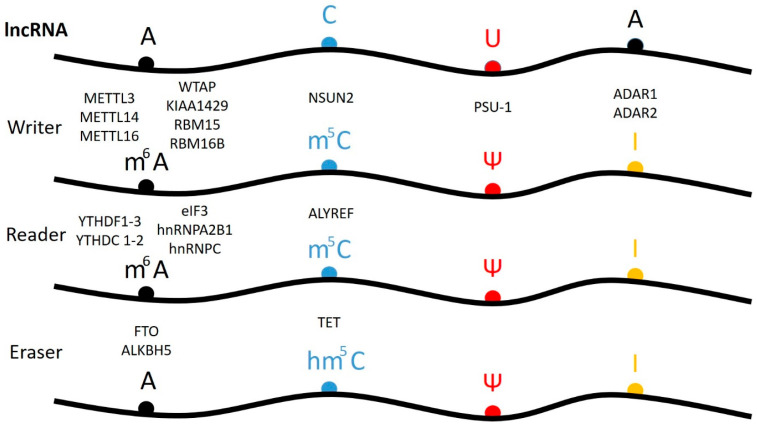
Schematic of lncRNA modification pathways. Writer, reader and eraser proteins are shown. **Nucleotides:** A—adenosine; C—cytidine; U—uridine. **Modifications**: m^6^A—6-methyladenosine; m^5^C—5-methylcytosine; Ψ—pseudouridine; I—inosine; hm^5^C—5-hydroxymethylcytidine. **Writer proteins**: METTL3, 14, 16 methyltransferase-like; WTAP—Wilms’ tumor 1 associating protein; KIAA1429—methyltransferase; RBM 15—RNA-binding motif protein 15 and its paralog RBM15B; NSUN—NOL1/NOP2/SUN domain family member; PUS—pseudouridine synthase; ADAR1, ADAR2—adenosine deaminase acting on RNA. **Reader proteins**: YTHDC, YTHDF—subgroups of YTH domain containing proteins; ZCCHC4—zinc-finger CCHC domain-containing protein 4; eIF3—eukaryotic initiation factor 3; hnRNPA2B1 and hnRNPC—heterogeneous nuclear ribonucleoproteins; ALYREF—ALY/REF export factor. **Eraser proteins**: FTO—fat mass and obesity-associated protein; ALKBH5—alpha-ketoglutarate-dependent dioxygenase AlkB homolog 5; TET—ten–eleven translocation protein.

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
