# Peer review of "Long Non-Coding RNA Epigenetics"

_ijms, 2021, doi:10.3390/ijms22116166_

Round 1
Reviewer 1 Report
The review manuscript entitled “Long non-coding RNA epigenetics” by Dr. Kazimierczyk and dr. Wrzesinski forgives a detailed overview of the current knowledge concerning the epigenetics of Long non-coding RNA. In particular they described the role of LncRNA methylation at several nucleotides belonging to these molecules, such as 6-methyladenosine, 5-methylcytidine, pseudouridine and inosine in regulating several cell pathways. TOverall, the text is interesting, well written and easy to follow. It will improve our knowledge regarding the posttranscriptional modifications of LncRNAs and their regulative role in a number of cellular pathways. As dysregulations in posttranscriptional modifications of LncRNAs can lead to disease, LncRNA epigenetics is an interesting research area which deserve attention.
Strengths
Well written, detailed and organized
Limitations
The quality of the figures should be improved
Few concepts are lacking in supporting references, I have suggested some refs
I have one main observation
LncRNAs play a role as signal, scaffold, guide, and decoy. These general lncRNAs mode of actions should be, at least briefly, described in the introductive section (PMID: 24965208), following the sentences about the lncRNAs biogenesis. It would be helpful for the reader to understand the general function of these RNA molecules.
Minor comments
Line7 “Long noncoding RNA ”should be “Long non-coding RNAs (lncRNAs)“
Lines 36-37. LncRNAs have also been found to regulate the osteogenic differentiation of Mesenchymal Stem Cells (PMID: 33898434). This information/ref should be included.
Line 76 ADAR enzymes should be reported as Adenosine deaminases acting on RNA (ADAR) when quoted the first time. It is also cited as “Adenosine deaminase” in line 223.
Lines 101-103 This sentence is lacking in references. For instance (PMID: 33795874)
Line 131 there is a period colored in red after “signaling”
Line 180 “understand.”--> understood?
Linme 231 “ADAR1 and 2 inactive” à It is unclear whether the author for “2” mean “ADAR2”
Lines 239-241 This sentence is lacking in references
Lines 264-267 a reference describing the first strategy for studying RNA modifications should be included
Lin 329 the references should inside square parenthesis
Line 399 It would be helpful for the reader to include the websites, if present, of these tools within the text
Line 401 EpiNano and Epinano should be uniformed
Lines 423—429 These sentences are lacking in references
Line 475 please include a period between “[173] “ and “Within”
Reviewer 2 Report
This review article reads well. I noticed only a few minor text changes to be revised.
Line 230 - delete "documented"
Line 236 - rephrase " appears to be a good" - it is unclear
Line 239 - 241 - reference is missing, or?
Line 259-260 - rephrase the sentence
Line 265 - delete "making it possible"
Line 308 - delete "makes it possible to" - detected; add a reference
Line 416 - delete "was performed" and change highlighting to highlighted
Line 432 - specify modifications
Line 451 - replace " at this stage" - so far/ to date
Line 477 - add reference
Author Response
Reviewer 2
Our responses to the reviewer's comments and suggestions are marked in blue
Line 230 - delete "documented"
Line 265 - delete "making it possible"
Line 416 - delete "was performed" and change highlighting to highlighted
Line 451 - replace " at this stage" - so far/ to date
As suggested by the reviewer, these changes were introduced into the manuscript.
Line 236 - rephrase " appears to be a good" - it is unclear
The phrase “The lncRNA-mRNA duplex appears to be a good substrate for the ADAR double stranded specific enzyme, which converts adenine to inosine.” is replaced as follow The lncRNA-mRNA duplex is recognized by the ADAR double stranded specific enzyme, which converts adenine to inosine.
Line 239 - 241 - reference is missing, or?
In position 117 we added new reference Silvestris, D.A.; Scopa, C.; Hanchi, S.; Locatelli, F.; Gallo, A. De Novo A-to-I RNA Editing Discovery in lncRNA. Cancers 2020, 12, 2959.
Line 259-260 - rephrase the sentence
The epigenetic mapping NGS methods currently used are based on NGS and as such, they typically do not detect modified nucleosides [117,118].
New version: Application NGS methods to epigenetic mapping is so far difficult because they typically do not detect modified nucleosides [117,118].
Line 308 - delete "makes it possible to" - detected; add a reference
The application of NGS after modification makes it possible to detect m6A sites in MALAT1 lncRNA.
New version The application of NGS after modification to detect m6A sites in MALAT1 lncRNA.
Line 432 - specify modifications
Due to its size, XIST has many modifications [26]
After correction Due to its size, XIST has many modifications like 78 sites m6A. 5 sites m5C and single site of Y [26].
Line 477 - add reference
Previously was “HOTAIR interacts with the nuclear m6A reader YTHDC1 at the methylated A783 and at additional sites”
We suggest add reference 180 which describe this observation
HOTAIR interacts with the nuclear m6A reader YTHDC1 at the methylated A783 and at additional sites [180].